# Optical Coherence Tomography Angiography in Intermediate and Late Age-Related Macular Degeneration: Review of Current Technical Aspects and Applications

**Adriano Carnevali** [1,*,†], **Rodolfo Mastropasqua** [2,†], **Valentina Gatti** [1], **Sabrina Vaccaro** [1], **Alessandra Mancini** [1], **Rossella D'Aloisio** [3,†], **Marco Lupidi** [4,5,6,†], **Alessio Cerquaglia** [4], **Riccardo Sacconi** [7,†], **Enrico Borrelli** [7,†], **Claudio Iovino** [8,†], **Livio Vitiello** [9], **Mario Damiano Toro** [10,11], **Aldo Vagge** [12,†], **Federico Bernabei** [13,†], **Marco Pellegrini** [13,†], **Antonio Di Zazzo** [14,†], **Matteo Forlini** [15,†] **and Giuseppe Giannaccare** [1,†]

1   Department of Ophthalmology, University Magna Graecia of Catanzaro, 88100 Catanzaro, Italy; valentina.gatti91@gmail.com (V.G.); sabrina_vaccaro@libero.it (S.V.); alessandra.mancini@studenti.unicz.it (A.M.); giuseppe.giannaccare@unicz.it (G.G.)
2   Ophthalmology Clinic, University of Modena and Reggio Emilia, Azienda Ospedaliero-Universitaria di Modena, 41125 Modena, Italy; rodolfo.mastropasqua@unimore.it
3   Ophthalmology Clinic, Department of Medicine and Science of Ageing, University G. D'Annunzio Chieti-Pescara, 66100 Chieti, Italy; rossella.daloisio@unich.it
4   Department of Surgical and Biomedical Sciences, Section of Ophthalmology, University of Perugia, S. Maria della Misericordia Hospital, 06156 Perugia, Italy; marco.lupidi@ospedale.perugia.it (M.L.); alessio.cerquaglia@studenti.unipg.it (A.C.)
5   Fondazione per la Macula Onlus, Di.N.O.G.Mi., University Eye Clinic, Viale Benedetto XV 5, 16132 Genova, Italy
6   Centre de l'Odéon, 113 Boulevard St. Germain, 75006 Paris, France
7   Department of Ophthalmology, University Vita-Salute, IRCCS Ospedale San Raffaele, Via Olgettina 60, 20132 Milan, Italy; sacconi.riccardo@hsr.it (R.S.); enrico.borrelli@hsr.it (E.B.)
8   Eye Clinic, Multidisciplinary Department of Medical, Surgical and Dental Sciences, University of Campania Luigi Vanvitelli, 80131 Naples, Italy; claudio.iovino1@unicampania.it
9   Eye Clinic, Department of Medicine, Surgery and Dentistry, "Scuola Medica Salernitana", University of Salerno, 84081 Salerno, Italy; lvitiello@unisa.it
10  Department of Ophthalmology, University Hospital of Zurich, University of Zurich, 9081 Zurich, Switzerland; toro.mario@email.it
11  Faculty of Medicine, Collegium Medicum, Cardinal Stefan Wyszyński University, 01815 Warsaw, Poland
12  Eye Clinic of Genoa, DiNOGMI—University of Genova IRCCS Ospedale, Policlinico San Martino, 16132 Genova, Italy; aldo.vagge@unige.it
13  Ophthalmology Unit, Department of Experimental, Diagnostic and Specialty Medicine (DIMES), Alma Mater Studiorum University of Bologna, S.Orsola-Malpighi Teaching Hospital, 40138 Bologna, Italy; federico.bernabei2@studio.unibo.it (F.B.); marco.pellegrini9@studio.unibo.it (M.P.)
14  Department of Ophthalmology, University Campus Bio-Medico of Rome, 00128 Rome, Italy; a.dizazzo@unicampus.it
15  Department of Ophthalmology, Istituto per la Sicurezza Sociale, San Marino State Hospital, 47890 San Marino, San Marino; matteo.forlini@iss.sm
*   Correspondence: adrianocarnevali@unicz.it; Tel.: +39-3339275424
†   Member of the Young Ophthalmologists Reviews Study Group (YORSG).

**Abstract:** Optical coherence tomography angiography (OCTA) is a non-invasive diagnostic instrument that has become indispensable for the management of age-related macular degeneration (AMD). OCTA allows quickly visualizing retinal and choroidal microvasculature, and in the last years, its use

has increased in clinical practice as well as for research into the pathophysiology of AMD. This review provides a discussion of new technology and application of OCTA in intermediate and late AMD.

**Keywords:** age-related macular degeneration; macular neovascularization; optical coherence tomography angiography (OCTA); retinal disease; posterior segment

## 1. Introduction

Age-related macular degeneration (AMD) is the third leading cause of severe irreversible vision loss worldwide, and it represents the major cause of central blindness in developed countries, especially among people older than 60 years [1–3]. Prevalence data suggest that about 200 million of people are nowadays affected by AMD, and this value is expected to increase to nearly 300 million by 2040 [4].

The advancement of imaging technology, in particular the use of optical coherence tomography (OCT) and OCT angiography (OCTA), has improved scientific knowledge on AMD, making mandatory a multimodal imaging approach for retinal and choroidal conditions. OCTA allows a clear and detailed visualization of retinal and choroidal microvasculature, and it is useful either for reaching the diagnosis or for guiding treatment choice and monitoring AMD patients [5,6].

In 2013, Ferris proposed a five-stage AMD clinical classification based on the risk of progression [7]. According to this classification, the presence of only small drusen ≤63 μm without pigmentary abnormalities is considered normal aging change; early AMD is characterized of medium drusen >63 μm and ≤125 μm, while intermediate AMD (iAMD) is defined with the presence of large drusen >125 μm and/or pigmentary abnormalities. Late AMD develops when macular neovascularization (MNV) or geographic atrophy (GA) occur [7].

MNVs are a growth of abnormal vessel and associated tissues into the outer retina, subretinal space, or subretinal pigmentary epithelium (RPE) space. They are classified according to the anatomic location determined by OCT imaging into three types: type 1 is a growth of vessels from choriocapillaris that proliferates into and within the sub-RPE space; type 2 originates from the choroid, and it passes through the Bruch's membrane and the RPE monolayer proliferating in the subretinal space; type 3 develops from the retinal circulation, usually in the deep capillary plexus, and it grows toward the outer retina [8,9].

MNV are usually characterized by intraretinal or subretinal fluid within the macula, but recent studies demonstrated that type 1 MNV can present without exudation on OCT, but it can be well visualized by means of fluorescein angiography (FA), indocyanine green angiography (ICGA), and OCTA [10,11]. The Consensus on Neovascular AMD Nomenclature (CONAN) group stated that this form of MNV could be identified more commonly with the improvement of imaging technology, but there is still not a consensus on which term to use [9]. In this article, we will use the term nonexudative MNV, even if there is a substantial difference between the term "treatment-naïve quiescent neovascularization" used by Querques and colleagues [10] and "asymptomatic, nonexudative or subclinical MNV" used by other authors [12–15]. Treatment-naïve quiescent neovascularization are MNVs without sign of activity for at least 6 months from baseline, while all the other terms refer to new diagnosis MNVs without exudation [10,12–15].

## 2. Materials and Methods: Search Strategy Design

A preliminary Pubmed search strategy was built up by an experienced information specialist, aided by all authors, combining free-text terms and synonyms of "optical coherence tomography angiography" and "age-related macular degeneration". Only scientific articles wrote in English language in the last 5 years were included for internal review and were further evaluated. Overall,

there were 2246 scientific articles, and after analyzing manuscript's titles and abstracts, 89 articles were selected. Moreover, a few select articles published before 2015 have been cited for historical purposes.

## 3. AMD Diagnosis

In the last decades, new multimodal imaging techniques have been incorporated to study AMD. They have led to a remarkable improvement in both understanding of the pathophysiology of macular diseases and its progression, but most importantly, to monitor treatment response [16].

FA is an irreplaceable dye-based invasive diagnostic tool to detect subtle neovascularization, monitor leaks, and compare them with staining and window defects. In the era of photodynamic therapy, FA has been considered the gold standard method for detecting and classifying subtle choroidal neovascularization as classic, occult, or combination subtypes [16,17]. Nowadays, it is still a useful method for the detection of "active" neovascularization and geographic atrophy [16]. However, fluorescein dye leaks from blood vessels, making it less ideal for visualization of details in the choroidal circulation [18]. Although FA is useful for the visualization of MNV, retinal–choroidal anastomoses and other choroidal vascular abnormalities are better visualized using ICGA [18]. In contrast to FA, it uses a dye that is 98% protein bound, providing more detailed images of the choroid and thus identifying the entire extension of the MNV [16]. However, both modalities have many drawbacks: they are time consuming and invasive, requiring intravenous dye injection, which can cause some side effects such as nausea and anaphylaxis [19].

In last two decades, OCT has emerged as a new non-invasive diagnostic tool for the diagnosis and follow-up treatment of macular diseases. Particularly in AMD, OCT facilitates in vivo high-resolution evaluation of the retina [20], so detecting the presence of drusen, RPE atrophy, fibrovascular complex, sub and intraretinal fluid, among other features [18].

More recently, OCTA has become available to retinal specialists. Unlike traditional angiography and ICGA, OCTA is a quick and non-invasive 3D imaging modality that does not require the use of a contrast agent [21]. OCTA detects the erythrocyte movement by analyzing the signal decorrelation within multiple B-scans performed repeatedly at the same location of the retina [12,16,21]. Changes in temporal contrast at a specific location indicate movement (erythrocyte motion) and hence vessel location [21].

OCTA allows the direct visualization and measurement of the foveal avascular zone (FAZ) area and provides morphologic and quantitative vascular information on macular microcirculation, including the deep and superficial capillary plexus, with good reproducibility and repeatability, in vivo and without dye leakage and staining that may obscure the limits and anatomy of pathologies [22].

However, as with any other imaging methods, several image artifacts also occur in OCTA, such as the shadow effect or those due to algorithms for data acquisition and image processing and motion-related artifacts [21].

## 4. Current OCTA Devices

OCTA represents an emerging non-invasive imaging technology that provides detailed visualization of the retinal and choroidal vascular networks. This technique employs the principle of motion contrast in order to generate blood flow and thereby images the vasculature without the need for a contrast agent [23]. Since its introduction, OCTA has allowed a deep characterization of several retinal pathologies, including AMD, diabetic retinopathy, vascular diseases, and also different inherited retinal diseases such as Stargardt macular dystrophy, Best disease, and choroideremia [24,25]. OCTA is able to compare the signals of sequential OCT B-scans from the same cross-section and distinguish the moving scatters from the tissue in the background in order to provide an image of retinal and choroidal networks [23]. In this way, it allows acquiring angiograms in a short time (<5 min), without the use of dye.

OCTA technology is constantly evolving to improve image quality, acquisition time, and the automatic interpretation of scans. Table 1 shows the comparison of different commercially available

OCTA devices: Zeiss AngioPlex, Cirrus HD-OCT 6000; Optovue AngioVue, RTVue XR AVANTI; Topcon Triton, DRI Triton; Heidelberg Spectralis, OCT2, Angiography; Nidek AngioScan RS-3000 Advance 2 and Canon Angio Xephilio OCT-S1. Table 1 compares the main characteristics provided by the manufacturer, including central wavelength, scanning speed, resolution, imaging depth, and imaging size.

**Table 1.** Different main features provided by the manufacturer of the different OCTA devices.

| OCTA System | Central Wavelength (nm) | Scanning Speed (Scans/s) | Resolution (Axial × Transverse, μm) | Imaging Depth (mm) | Imaging Size (mm) | OCTA Approach |
|---|---|---|---|---|---|---|
| Zeiss AngioPlex, Cirrus HD-OCT 6000 | 840 | 100,000 | 5 × 15 | 2.0–2.9 | 3 × 3, 6 × 6, 8 × 8, 12 × 12 | Combined intensity and phase variance |
| Optovue AngioVue, RTVue XR AVANTI | 840 | 70,000 | 5 × 15 | 2.0–3.0 | 3 × 3, 6 × 6, 8 × 8 | Intensity decorrelation |
| Topcon Triton, DRI Triton | 1050 | 100,000 | 8 × 20 | 2.6 | 3 × 3, 6 × 6 | Intensity ratio analysis |
| Heidelberg Spectralis, OCT2, Angiography | 870 | 85,000 | 5 × 6 | 2 | 3 × 3 | Intensity decorrelation |
| Nidek AngioScan RS-3000 Advance 2 | 880 | 85,000 | 7 × 20 | 2.1 | 3 × 3, 4.5 × 4.5, 6 × 6, 9 × 9 | Combined intensity and phase decorrelation |
| Canon Angio, Xephilio OCT-S1 | 855 | 100,000 | NA | 5.3 | 3 × 3, 6 × 6, 10 × 10, 20 × 23 | NA |

The OCTA devices available at the present can do 70,000 to 100,000 scans per seconds. The scanning speed is related to the presence of motion artifact and to the image resolution. In particular, the higher the scanning speed, the lower the motion artifacts that would be present in the final image. This feature plays a fundamental role in the clinical practice, where it is often necessary to optimize exam times. Recently, Topcon, Zeiss, and Canon developed novel systems that allow a scanning speed of 100,000 scans per second.

In addition, imaging depth has been improved compared to the first models. In particular, in current available devices, the imaging depth ranges from 2.0 to 5.3 mm.

The software for the visualization of volumetric data, and segmentation algorithms, is different between the devices and hardly comparable. Different approaches, such as the amplitude decorrelation algorithm and the combined intensity and phase decorrelation, have been developed to improve image quality and to reduce the motion artifacts [26–28].

Finally, the image size has been significantly increased over the years. In fact, in the first models, only 3 × 3 or 6 × 6 scans were possible. Recently, 12 × 12 by Zeiss and even 20 × 23 by Canon have also been introduced.

Moreover, two main Fourier domain detection systems of OCTA are available: the spectral domain (SD-OCTA) and the swept source OCTA (SS-OCTA) [16].

The SD-OCT employs a broadband near-infrared superluminescent diode that has a wavelength of 840 nm, with a spectrometer to measure wavelengths of light. The SS-OCT instead employs a tunable swept laser that has a wavelength of 1050 nm and uses a single photodiode detector [16,29,30].

The main advantages of SS-OCTA imaging over SD-OCTA are (i) the faster scanning speed, which allows for denser scan patterns and larger scan areas compared with SD-OCT scans for a given acquisition time, it presents a faster scanning speed, resulting in a higher number of scans for a given

acquisition time compared with SD-OCT, (ii) its reduced sensitivity roll-off, resulting in enhanced light penetration through the RPE, as well as a better detection of signals from the sub- RPE layer due to its reduced sensitivity roll-off, which allows a better light penetration through the RPE and thus a greater detection of signals from the sub-RPE layer [31], which is particularly important in case of drusen or RPE thickening (iii). Since that resolution depends mainly on the wavelength, increasing this parameter allowed a better axial resolution, resulting in a better characterization of the different layers even in the presence of obstacles such as the presence of cataracts or vitreous opacities.

The higher wavelength in combination with the reduced sensitivity roll-off improves the detection the weaker signals from the deeper layer, resulting in a better detection of type 1 MNV compared with SD-OCTA imaging [30,31].

## 5. OCTA in Intermediate AMD

AMD may present at different stages, and the "intermediate AMD" stage is clinically characterized by the presence of pigmentary abnormalities and/or large drusen [7]. Of note, eyes with subretinal fluid may be also characterized by the absence of neovascularization and thus classified as iAMD [32].

Although the AMD pathogenesis is intricated and related to several systemic and lifestyle factors that may have a role in the development and progression of this disorder [33,34], a growing body of evidence suggests that this disorder is ultimately characterized by damage of the unit comprised of photoreceptors, RPE, Bruch's membrane, and choriocapillaris (CC) complex [35–37]. Importantly, several pieces of evidence suggest that this may be considered as a tightly knit, integrated unit [38–40]. In AMD, this impairment causes the development of drusen and progressive photoreceptor, RPE, and CC degeneration [41–44].

Previous studies have fully characterized the CC perfusion in eyes with iAMD [25,45–48]. In detail, using both spectral domain and swept source technologies, Borrelli and colleagues demonstrated that the CC is impaired (i.e., CC ischemia) in these eyes [45,46]. Importantly, they provided a topographical analysis revealing that the CC impairment is mainly confined to the CC beneath and surrounding drusen [46]. Moreover, the CC perfusion was more affected in iAMD eyes with neovascular AMD in the fellow eyes [45], especially in those with type 3 MNV [47]. In two studies employing OCTA in iAMD eyes [49,50], the authors showed that the CC perfusion is strictly correlated with macular function in these eyes, further highlighting the association between CC ischemia and outer retina dysfunction in iAMD.

Although the CC is known to be the vascular plexus most affected in AMD, also retinal vessels were demonstrated to be impaired in these eyes [51,52]. Toto et al. demonstrated that eyes with iAMD are characterized by a lower retinal perfusion as compared with normal eyes [51]. Of note, iAMD eyes with OCT signs of nascent GA are featured by a greater reduction in retinal vessels perfusion [52].

## 6. OCTA in Type 1 MNV Secondary to AMD

Type MNV has been identified as the main complication of AMD [53]. The typical "pin-points" hyperfluorescent spots at the late stages of FA and the plaque pattern in ICGA characterize type 1 MNV lesion that peculiarly grows under RPE [54]. Different responses to anti-vascular endothelial growth factor (VEGF) treatment can be observed with a consequent variable visual outcome of subjects suffering from such a condition.

To date, many advances in multimodal imaging have been made and routinely added in the clinical practice to better identify and follow MNV lesions, studying in detail retinal and choroidal microvasculature combined with the co-registered OCT structural scan.

Given the longer wavelength and the wider scan, SS-OCTA imaging modality provides a more qualitative analysis of tissues beneath the RPE, thus allowing a precise evaluation of perfusion and structural features, despite drusen-related shadowing artifacts [30].

SS-OCTA has also the advantage of providing a full extension of MNV lesion, if compared with spectral domain imaging modalities and is beneficial particularly in asymptomatic eyes with presumed intermediate AMD, given the systemic risks of FA/ICGA as well as their costs.

Furthermore, the detection of subclinical MNV using OCTA is being crucial to identify early signs of the disease activity and to investigate some biomarkers that have been considered as predictors in terms of functional and anatomical outcomes. Indeed, subclinical MNV can appear as slightly elevated RPE with moderately reflective material in the sub-RPE space. Shi et al. have considered the OCT double-layer sign as a predictive biomarker in identifying subclinical neovascularization in AMD [55]. Double-layer sign has been observed on OCT scan as an irregular pigment epithelial detachment described as highly reflective layers corresponded to a little separation between the RPE and another highly reflective layer and Bruch's membrane [55].

Some OCTA-based criteria have been proposed as well, especially to identify different subgroups [56].

Using advanced post-processing analysis, Arrigo et al. described an OCTA-based classification of type 1 MNV, quantifying OCTA features in 120 patients with newly diagnosed type 1 MNV and distinguishing two different subgroups: a high vessel tortuosity (VT) subgroup and a low VT one [57]. The latter seems to have a better clinical prognosis with higher final visual outcomes and with lower signs of degenerative outer retinal alterations. On the other hand, the patients presenting high VT have a more florid neovascular network and probably more aggressive nature (Figure 1).

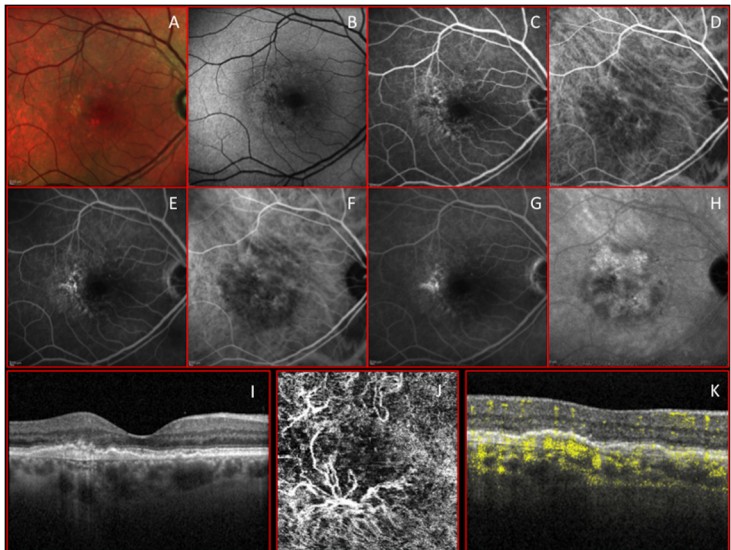

**Figure 1.** Multimodal imaging clearly shows a treated type 1 macular neovascularization (MNV). (**A**,**B**) Color and autofluorescence images; (**C**–**E**) early, middle, and late phase of fluorescein angiography. (**F**–**H**) Early, middle, and late stage of indocyanine green. (**I**–**K**) 3 mm × 3 mm optical coherence tomography angiography (OCTA) and co-registered structural optical coherence tomography (OCT) scan show that a florid arborization of vessels is evident at the outer retina slab with a high choroidal flow void signal under the subretinal pigmentary epithelium (RPE) in correspondence with the macular neovascularization (MNV) lesion.

Although the presence of the intraretinal hyper-reflective foci is a known biomarker of the MNV's response to treatment, it seems to be less reliable than VT in differentiating clinically relevant MNV subgroups, suggesting a key role in VT as a predictive biomarker of lesion worsening with the onset of subretinal fibrosis. However, the authors noticed that both subgroups (high and low VT) required a similar number of intravitreal injections.

Of note, Farecki et al. observed that type 1 MNV has fewer sharp boundaries with a larger extension if compared with a type 2 MNV on an OCTA scan [58].

The OCTA categorization of type 1 MNV secondary to AMD has become essential both in research field and in the clinical setting to have a better knowledge about the prognosis and response to treatment.

Pilotto et al. described early MNV changes after intravitreal anti-VEGF using OCTA imaging, such as a decrease in MNV mean area after treatment clearly visible at en face OCTA images and reduction in pigment epithelial detachment (PED) height and central macular thickness (CMT) on an OCT scan [59]. Only after 48 h, in most cases, a decreased visibility of the littler choroidal branching vessels was detected, which was not always associated with a persistence of flow from larger vessel trunks. Similarly, Spaide et al. had already observed the behavior of new vessels during anti-VEGF treatment, providing qualitative and quantitative changes in the morphology and flow of the network [60].

A reduction of VEGF after injections seems to lead vessel vasoconstriction combined with the flow signal regression [61].

One of the main limitations of OCTA in investigating type 1 MNV remains the lack of subtle leakage identification, which is easily seen from the traditional FA. Nevertheless, the traditional FA combined with ICGA is really time consuming and requires the use of dye with possible systemic complications. Farecki et al. have reported that FA does not give more information for the management of exudative AMD if compared with OCT imaging [58].

Another important OCTA limitation that deserves mentioning is represented by a projection artifact that in AMD conditions is from inner retinal vessels to both above and under RPE. To solve projections-related problems, the masking technique was firstly proposed blocking the larger vessel projection of the en face superficial retinal slab to the en face outer retinal and choriocapillaris slabs [62]. Unfortunately, this method removed the flow signal from blood vessels in the outer retinal layer as well.

Another projection removal system proposed was the subtraction of flow signal of superficial vessels from flow signal detected in the deeper retinal layers, thus allowing a better exploration of MNV extension. Nevertheless, it also caused the flow signal attenuation of the choroidal lesions [63,64].

A new algorithm for projection-resolved OCTA (PR-OCTA) with a high within-visit repeatability has been developed, acting on the single voxels, thus providing more resolution of MNV on both en face and cross-sectional images [65–67].

The current standard of classifying MNV type seems to remain a multimodal imaging approach that utilizes both FA/ICGA and OCTA combined with the co-registered structural OCT. However, OCTA can give a new understanding of the vascular features and changes over time and/or after treatment of macular new vessels and has become a fast imaging tool in clinical real-life settings for the follow up and management of exudative macular lesions.

## 7. OCTA in Type 2 MNV Secondary to AMD

Type 2 neovascular networks in AMD originate from the choroid and are located in the subretinal space [8]. The presence of subretinal hyperreflective material (SHRM) on structural OCT is a common finding in type 2 MNVs, and it was reported to be highly correlated with visual function worsening [68]. SHRM was described to be related to several components as neovascular tissue, hemorrhage, subretinal exudation (Figure 2), and fibrosis [69,70]. Sometimes, it might be challenging to distinguish between SHRM components using conventional imaging; therefore, OCTA can be decisive in distinguishing the neovascular from the exudative component [71]. Kuehlewein et al. described the visualization of a large type 2 neovascular lesion on OCT angiograms [72]. Two large central trunks and a dense branching network of smaller caliber vessels radiating in all directions from the main trunk were appreciated [72]. El Ameen et al. observed different type 2 MNV patterns on OCTA: the "medusa-shaped" and the "glomerulus-shaped" [73]. In the medusa-shaped pattern, a compact zone of tiny blood vessels with a minimal hypodense structure inside was present, whereas the glomerulus-shaped lesions were compared with the kidney glomerulus as globular structures of entwined vessels separated by hypodense spaces [73]. The capability of OCTA in detecting the flow

signal at the level of SHRM was demonstrated even when type 1 and 2 components coexist in the same neovascularization, the so-called mixed lesion [74]. Coscas et al. observed on OCT angiograms a qualitative and quantitative reduction of both type 1 and type 2 components of vascular networks after intravitreal VEGF-trap treatment [74]. Dolz-Marco et al. described the regression of type 2 lesions into a type 1 pattern after anti-VEGF treatment [75]. OCTA was also able to show the disappearance of flow signal in the subretinal space with the progressive attenuation and dislocation beneath the RPE of the same flow signal after treatment [75].

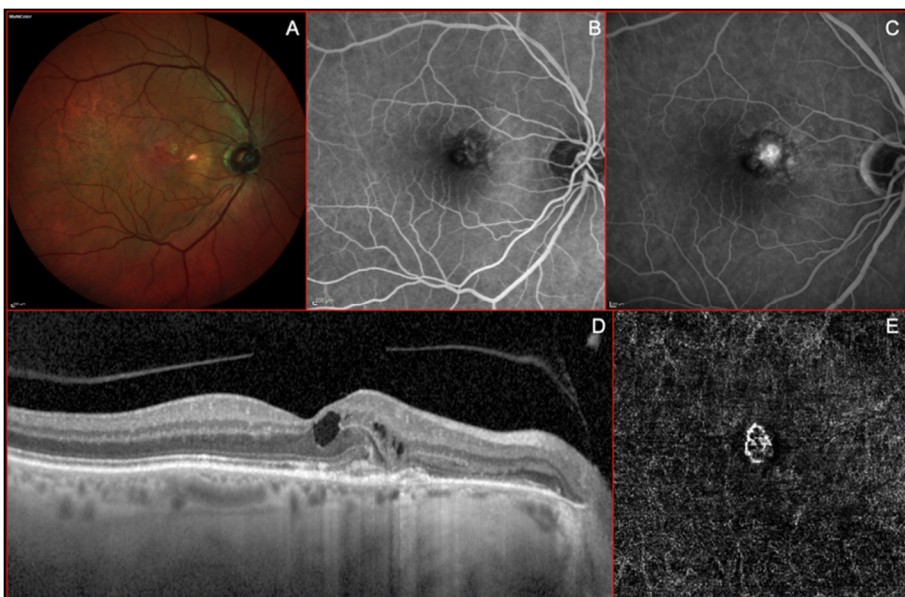

**Figure 2.** Imaging of a Type II or "classic" macular neovascularization (MNV). (**A**) Multicolor imaging showing some reddish lesion in macular area due to pigmentary changes associated with a yellowish area secondary to a focally increased reflectivity at the level of the outer retina. (**B**) Early venous phases of fluorescein angiography (FA) showing a well-defined neovascular network in the nasal side of macular area with a clear dye-leakage during the late frames (**C**). (**D**) Structural optical coherence tomography (OCT) showing a well-defined subretinal mild hyperreflective lesion, located nasally to the foveal depression, associated with intraretinal fluid accumulation (cystoid spaces). (**E**) A fine neovascular network is visible at the level of the outer retina on "en-face" OCT-angiogram.

Moreover, the OCTA detection of neovascular networks in patients with subretinal fibrosis secondary to neovascular AMD was reported to be associated with poorer visual function outcomes [76]. Several authors have quantitatively analyzed OCT-A biomarkers of both type 1 and type 2 MNVs, detecting different behaviors of neovascular networks [62,72,74,77,78]. Jia et al. reported that larger lesions and type 2 MNVs showed higher flow index, which is a parameter that was directly correlated with the presence of active blood flow within the vascular complex [62]. Zhao et al. observed that type 2 lesions showed a smaller flow area, smaller greatest vascular caliber (GVC), and smaller greatest linear dimension (GLD) when compared to type 1 lesions [77]. Moreover, type 2 networks were associated with a shorter duration of the disease, which is a parameter that was positively correlated with GVC [77]. Kuehlewein et al. observed that MNV vessel and lesion size reduction after anti-VEGF treatment was mostly prominent in type 2 MNVs, rather than in more complex and long-standing vascular networks as type 1 MNVs [72]. Even other parameters such as MNV area and GLD, computed on "en face" OCT angiograms, were reported to decrease after anti-VEGF treatment more in type 2 than in type 1 lesions [56,72]. Two new quantitative parameters have been recently proposed: the VT, which reflects the geometrical properties of a neovascular network, and the vessel dispersion (VDisp), which express the disorganization degree of the MNV [78]. Type 2 lesions seemed to show highest VDisp values than type 1 MNVs, whereas VT values were similar in both neovascular subtypes [56].

## 8. OCTA in Type 3 MNV Secondary to AMD

Type 3 MNV, also known as retinal angiomatous proliferation (RAP), is considered a peculiar form of neovascular AMD [79]. It originates from retinal vessels in the deep vascular complex and infiltrates the sub-RPE space [80]. Multimodal imaging evaluation including color fundus photography, FA, ICGA, SD-OCT, and OCTA is helpful in type 3 MNV diagnosis [80–82].

On SD-OCT, the lesion is characterized by the presence of a characteristic intraretinal hyperreflective lesion emanating from the deep capillary plexus (DCP) at the junction of the inner nuclear layer (INL) and outer plexiform layer (OPL) [81]. It is associated with intraretinal cystoid edema with or without subretinal RPE fluid [81]. The pathophysiology of type 3 MNV remains unclear, but in recent years, OCTA has increasingly played a greater role in the diagnosis and treatment monitoring of this retinal disease [83].

Concerning the diagnosis of type 3 MNV, OCTA has been shown to clearly display the retinal–retinal anastomosis [80]. Such lesions originate from the DCP, creating a clear, tuft-shaped, high-flow network in the outer retinal segment, abutting in the sub-RPE space [84] (Figure 3).

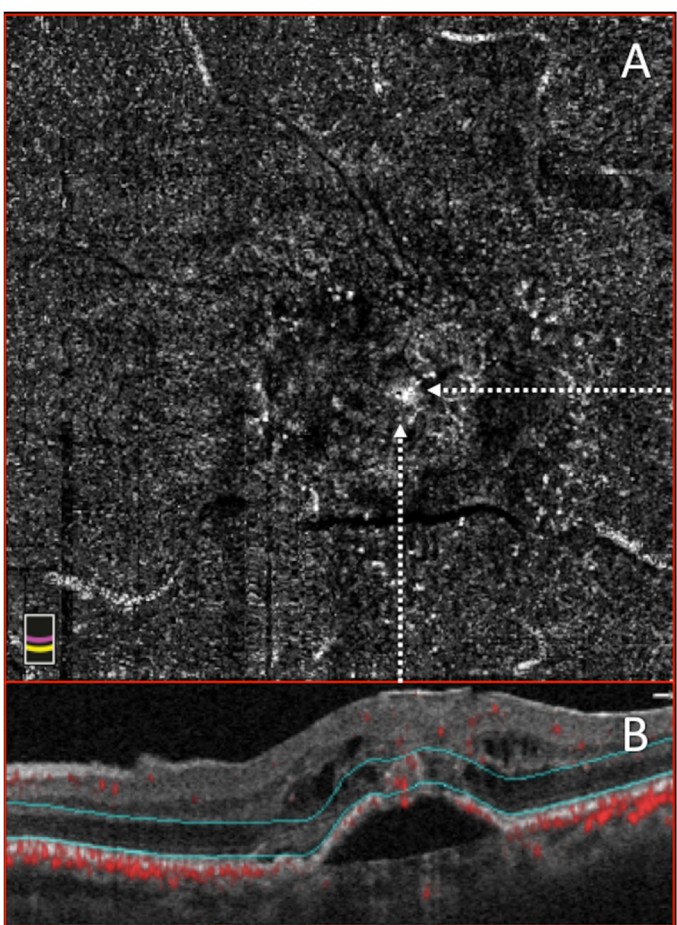

**Figure 3.** Optical Coherence tomography angiography (OCTA) and structural optical coherence tomography (OCT) of a Type III macular MNV. (**A**) 3 × 3 OCTA of outer retinal segment showing the presence of a white "point like" that corresponds to a detectable flow. (**B**) Co-registered structural OCT scan with hyporeflective cyst and hyperreflective intraretinal lesion.

Furthermore, the choriocapillaris segmentation slab could contain a small, clew-like lesion, connected in some cases with the choroid through a small-caliber vessel [84–86].

It is also possible to find the presence of intraretinal hyperreflective foci at the site of type 3 MNV development, which is characterized by detectable flow on OCTA, identified within the avascular

slab or either the DCP or avascular slabs [87]. Typically, these hyperreflective foci on OCTA are characterized by the absence of intraretinal exudation, or very mild microcystic changes, until the lesion progresses from the DCP into the RPE and sub-RPE space [87–89].

Using OCTA findings, a recent classification of type 3 MNV was also proposed, providing a better characterization of the disease [90]. Stage 1 is characterized by telangiectatic flow in the DCP without OPL disruption, stage 2 presents downward intraretinal flow and subretinal flow without RPE disruption, while stage 3 shows downward flow and RPE disruption [90].

Interestingly, OCTA evaluation of the fellow eye of patients with unilateral type 3 MNV also showed several significant changes. In particular, affected eyes showed increased choriocapillaris nonperfusion versus contralateral non-neovascular eyes, suggesting that choroidal ischemia could play an important role in the development of type 3 MNV [47]. In addition, contralateral eyes of patients with unilateral lesion show a reduced vascular perfusion compared to control eyes [91].

Of note, OCTA has been shown to be very useful in the type 3 MNV assessment after anti-VEGF therapy [92]. It clearly shows how the tuft-shaped abnormal outer retinal lesion tends to change after 1 year of anti-VEGF therapy. Either the flow becomes undetectable, or a sub-RPE neovascularization, with persistence of the DCP abnormalities, can develop [92].

Moreover, at the non-exudative stage after the treatment, OCTA can show a decrease of the flow inside the retinal lesions, with a regression of the connection between DCP and the RPE/sub-RPE space [93]. Whereas, at the time of recurrence of the exudation after the treatment, OCTA shows the presence of intra/sub-retinal exudation with restoration of the flow deepening from the DCP to the RPE/sub-RPE space [93].

OCTA has also confirmed the efficacy of a combined treatment with photodynamic therapy and anti-VEGF injections, showing a resolution of vascular and structural abnormalities, although this does not always correspond to a visual acuity improvement [94,95].

Overall, OCTA showed a good level of sensitivity and accuracy in the diagnosis and characterization of type 3 MNV by increasing the detection rate for these lesions and offering new insights into this retinal disease [96,97].

## 9. OCTA in Geographic Atrophy

Geographic atrophy (GA) represents the advanced form of dry AMD, and it affects more than 5 million people worldwide [4]. GA is characterized by the degeneration of photoreceptor cells, RPE, and CC. Usually, in the early stages of the disease, it involves the extrafoveal region and then includes the fovea, limiting daily activities and impairing quality of life [98–101].

GA is commonly assessed by color fundus photography and fundus autofluorescence (FAF), which is considered the gold standard for the evaluation of progression of atrophy enlargement [102–104].

Recently, several studies conducted using OCTA showed that the atrophic patches present a loss of choriocapillaris flow and an improved visualization of choroidal vessels [105]. In particular, GA appears with loss or rarefaction of CC at the level of RPE atrophy [105,106]. In these areas of CC impairment, large choroidal vessels may be displaced and may be seen on the en face OCTA image at the depth level where CC is ordinarily seen [105]. Corbelli et al. demonstrated that OCTA is an effective technique for the visualization and quantification of GA lesions [107]. Evaluating 47 eyes affected by GA, they showed that OCTA can quantify the area of atrophy as FAF and en face OCT [107].

Another advantage of OCTA is the ability to evaluate the CC status of eyes with GA. Sacconi et al. firstly reported a quantitative CC impairment surrounding the GA margin before RPE alterations observed using FAF, assuming a "*primum movens*" at the level of the CC [37]. Moreover, the same group and others showed a greater CC impairment in the area that subsequently developed GA expansion, suggesting that the impairment of CC flow could predict the enlargement of the atrophic lesion [24,108,109]. For this reason, OCTA showed that the CC impairment could be considered as a new a risk factor for GA progression and a biomarker to be measured to determine the efficacy of new interventions aiming to slow the progression of GA.

Finally, OCTA results useful in the evaluation of patients affected by GA in order to exclude the presence of treatment-naïve nonexudative MNV. Capuano et al. firstly reported the presence of nonexudative MNV in patients affected by GA, showing outcomes and follow-up of 19 eyes [110]. In detail, the authors reported a rate of activation of 26% during a follow-up of 45.7 ± 14.7 months. In this way, OCTA play an important role in the diagnosis and follow-up of patients affected by nonexudative MNV and GA.

In conclusion, the benefits of using OCTA compared with other imaging approaches for GA include the convenience of using only one type imaging technique for showing en face flow images and structural OCT data, the quantification of GA area, and the potential ability to exclude the presence of MNV without performing fluorescein angiography, particularly in treatment-naïve nonexudative type 1 MNV.

## 10. OCTA in Nonexudative MNV in AMD

Previous studies have demonstrated that MNV can present without clinical sign of exudation. In the 1970s, histopathological studies pointed out the presence of what is now known as type 1 MNV in patients without any hemorrhage or exudation [111,112]; later, Chang et al. reported that ICGA was able to detect subclinical MNV not viewable with FA, by the presence of hyperfluorescent plaque with late staining, in the fellow eyes harboring soft drusen [113]. Querques et al. detected on FA and ICGA type 1 MNV that had not developed intraretinal or subretinal exudation visualizable on OCT for at least 6 months, and they called this form treatment-naïve quiescent neovascularization [10]. In 2016, Roisman et al. observed in patients with previously diagnosed neovascular AMD in one eye and asymptomatic iAMD in the contralateral eye the presence on ICGA of MNV without exudation and subsequently confirmed by OCTA [12].

Nonexudative MNV can be found both in iAMD and GA [12–14,30,114]. Nonexudative MNVs were described also in other retinal diseases such as pachychoroid, large colloid drusens, and angioid streaks [115–117]. Carnevali et al. determined that OCTA has a good sensitivity and specificity for nonexudative MNV detection in eyes with iAMD, thus representing a valid diagnostic tool [114]. Several studies compared the ability of detecting MNV of SD-OCTA and SS-OCTA demonstrated that SS-OCTA provides better image quality and a more accurate representation of MNV than SD-OCTA [30,31].

In iAMD, nonexudative MNVs usually present on OCTA as irregular, without visible-core, with a well-defined margin and foveal-sparing vessel networks, even if sometimes, they do not present a detectable core vessel, and it was hypothesized that this could represent a protective factor against increased activity [113] (Figure 4).

They are usually located in the subfoveal/perifoveal area [114]. As mentioned above, Capuano et al. described nonexudative MNV in GA using OCTA, and this MNV showed different characteristics from those in iAMD [110]. OCTA in GA has a lower detection rate than in iAMD, which is probably because small MNV are more difficult to differentiate from choroid due to the low contrast between MNV and choroidal vessels in the absence of RPE and CC or due to their slow flow [110]. Nonexudative MNVs in GA show irregular-shaped high flow with well-defined margin networks without a visible core, and they present vessel arteriolization with few or without visible fine capillaries; they are usually found bordering the atrophic area [110].

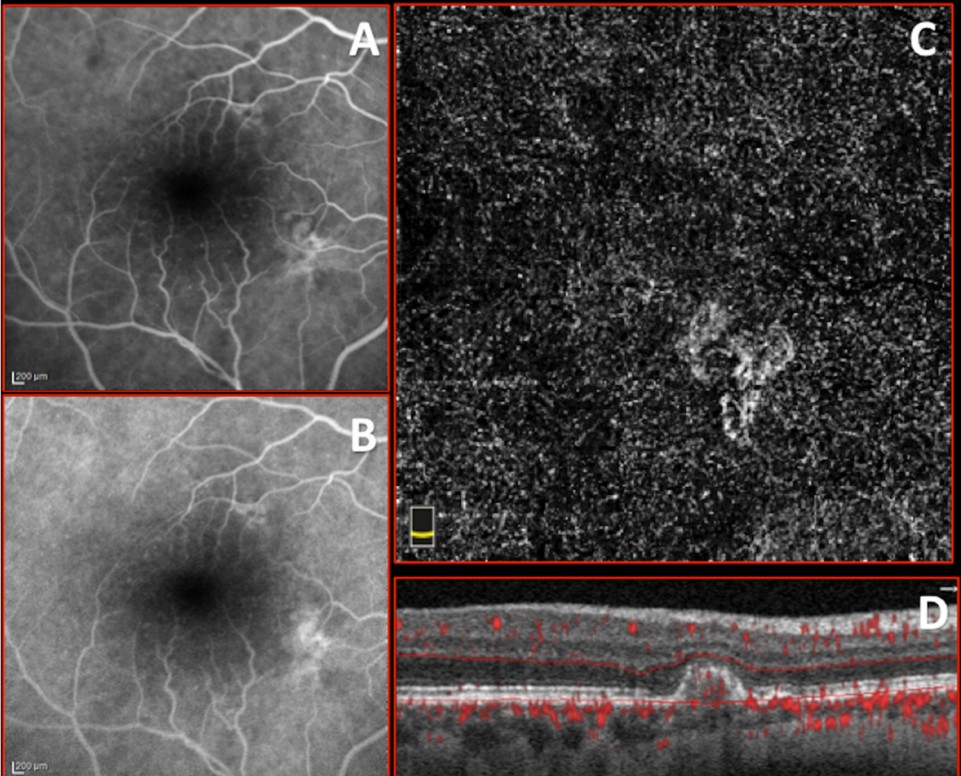

**Figure 4.** Imaging of treatment-naïve quiescent macular neovascularization (MNV). (**A,B**) Early and late stage of fluorescein angiography (FA) showing inhomogeneous small hyperfluorescence without leakage. (**C,D**) 3 × 3 optical coherence tomography angiography (OCTA) and corresponding structural OCT b scan of choriocapillary plexus showing circular, well defined, foveal-sparing treatment-naïve quiescent MNV with small loops.

The prevalence of nonexudative MNV was evaluated with OCTA by different groups, particularly in the contralateral eyes of patients with unilateral exudative AMD [13–15]. De Oliveira Dias et al. [13] and Yang et al. [14] reported a prevalence of 14.4% and 13.2% respectively at baseline of eyes with iAMD and GA, while Bailey et al. [15] found a prevalence of 7.9%. Capuano et al. analyzed 644 eyes from 399 consecutive patients with unilateral or bilateral GA secondary to AMD and identified 73 eyes from 71 patients (11%) with nonexudative MNV [110].

De Oliveira Dias et al. and Yang et al. found a cumulative incidence of exudation of 24% after 1 year and of 34.5% after 2 years of follow-up in eyes with iAMD [13,14], whereas Heiferman et al. described an incidence of exudation of 20% after 15 months of follow up in patients with active neovascular AMD in the contralateral eye [116]. Different results were obtained by Bailey et al., who found an incidence of exudation of 80% at 2 years [15]. These differences could be explained by several factors including different populations, the number of patients evaluated, and the duration of AMD. Carnevali et al. [11] and Capuano et al. [110] reported the rate of activation, following the criteria for diagnosis of quiescent CNVs, respectively of 6.6% in patients with iAMD at 1 year of follow-up and of 26.3% in eyes with GA within a mean of 20 months of follow-up.

OCTA may be useful to predict exudation; in fact, there was a reported increase in the size and area of the vascular network of nonexudative MNVs that later developed signs of activity [11,15,109,118,119]. Capuano et al. also described a morphologic change such as the sprouting of new vascular loops and tiny capillaries before exudation [110]. It was not found a significant difference between the area of the CC nonperfusion in eyes with exudative MNV and nonexudative MNV [120,121].

In conclusion, OCTA is a valid diagnostic tool for the identification of nonexudative MNVs in eyes with iAMD or GA for the evaluation of signs of exudation; it can be also used to predict a possible activation of nonexudative MNV to guide the time of follow-up and treatment.

## 11. Conclusions and Future Perspectives

In recent years, OCTA has become indispensable in clinical practice for both the diagnosis and follow-up of AMD as well as to evaluate the response to treatment with anti-VEGF agents. OCTA is a quick and non-invasive diagnostic tool that does not require the use of a contrast agent. It helps to identify early signs of AMD; it shows a good level of sensitivity and accuracy in the diagnosis of late stage of AMD; it also provides the correct characterization of subtypes of MNV, improving the detection rate and offering new insights into the pathophysiology and progression of these lesions.

OCTA provides some biomarkers as predictors of functional and anatomical outcomes improving patient management and therapy. Moreover, OCTA has several advantages, including a non-invasive and rapid acquisition of angiograms, depth resolution, and perfect microvascular resolution. On the contrary, the main limitation of current OCTA devices are image artifacts; however, OCTA technology is evolving in order to improve image quality, acquisition time, and automatic interpretation of scans, allowing for improved diagnosis and understanding of AMD [122].

Visualization of the choroid and choriocapillaris may be affected by the loss of flow signal with depth in short-wavelength OCTA devices, but this limitation may be addressed in the near future by new-generation of OCTA devices. In particular, artifacts of masking and unmasking are very common in patients with AMD. These regions are lined with areas of PEDs or large retinal vessels anterior to the choroid on B scans. The creation of artifacts due to PEDs are accompanied by focal unmasking artifacts due to RPE atrophy. In the region immediately below RPE atrophy, there is an increase in OCT reflectivity throughout the choroid. This is also accompanied by an augmented artifacts decorrelation signal in the underlying sub-levels.

Despite these disadvantages, OCTA is a promising imaging modality that may provide various precious information in patients with AMD, and it may help in correlating functional parameters. Moreover, OCTA might provide deeper knowledge in the pathophysiology of human choroid AMD and possibly allow predicting the natural history of the disease and choosing the best therapeutic approach. In particular, non-invasive in vivo identification of the histopathologic substrate and understanding of biogenesis of first signs of the disease will open the door to further therapeutic targets.

**Author Contributions:** A.C. (Adriano Carnevali): project administration, methodology, resources, writing—original draft preparation, writing—review and editing, supervision, visualization, data curation, formal analysis, R.M. and G.G.: project administration, methodology, formal analysis, investigation, writing—original draft preparation, writing—review and editing, supervision, visualization, V.G., S.V., A.M., R.D., M.L., A.C. (Alessio Cerquaglia), R.S., E.B., C.I., L.V., M.D.T., A.V., F.B., M.P., A.D.Z. and M.F.: writing—original draft preparation, writing—review and editing. All authors have read and agreed to the published version of the manuscript.

**Funding:** This review received no external funding.

**Conflicts of Interest:** The authors declare no conflict of interest.

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
