# Peer review of "Optical Coherence Tomography Angiography in Intermediate and Late Age-Related Macular Degeneration: Review of Current Technical Aspects and Applications"

_applsci, doi:10.3390/app10248865_

Round 1

Reviewer 1 Report

OCT in AMD review

  1. General remark: it is a valuable paper providing current knowledge on OCTA technique used for AMD diagnosis (a lot of work really), however it does not exactly have a character of a review. It is more like a book chapter on the subject. If authors want to provide a review,  they should refer precisely to the search in database and present different opinions on selected questions. I would suggest to change the category of this paper or reorganise it a bit.
  2. Table 1. How were the devices selected to be included in this table ? Please mention how was your choice made.
  3. In material and methods section authors provided search criteria. What are the results of the search, in detail ? Please provide this information.
  4. Paragraph 5 and the following paragraphs : I would start the paragraph from listing the most important aspects/problems in OCTA diagnosis of iAMD and than follow with review of literature referring to that. This kind of “pattern” I would suggest to use in all paragraphs: this way you will have a form of a review, not an essay.
  5. Moderate English editing and spell-check required.

Author Response

Corresponding Author:

Dr Adriano Carnevali, MD

Telephone: +39 09613647041

Email : adrianocarnevali@live.it

Dear Editor,

Thank you for considering our revised manuscript entitled Optical Coherence Tomography Angiography in Intermediate and Late Age-Related Macular Degeneration: Review of Current Technical Aspects and Applicationsfor publication in Applied Sciences.

The points raised by the Reviewers have been favorably taken into account and incorporated into the revised manuscript. We provide a point-by-point response to each of their comments below.

Best regards,

Adriano Carnevali

Comments from the editors and reviewers:

Reviewer #1

General remark: it is a valuable paper providing current knowledge on OCTA technique used for AMD diagnosis (a lot of work really), however it does not exactly have a character of a review. It is more like a book chapter on the subject. If authors want to provide a review, they should refer precisely to the search in database and present different opinions on selected questions. I would suggest to change the category of this paper or reorganise it a bit.

Response: We agree with the Reviewer and therefore we modified the section “Materials and Methods” improving details about the results of “search in database”: “A preliminary Pubmed search strategy was built up by an experienced information specialist, aided by all authors, combining free-text terms and synonyms of: “Optical coherence tomography angiography” and “age-related macular degeneration”. Only scientific articles wrote in English language in the last 5 years were included for internal review and were further evaluated. Of overall 2246 scientific articles, after analyzing manuscript′s titles and abstracts, 89 articles were selected. Moreover, few select articles published before 2015 have been cited for historical purposes.”

However, it should be highlighted that this is a descriptive review rather than a systematic review/meta-analysis.(page 2-3, line 93-160).

In order to address the request of adding comments to the data presented, we modified the section “Conclusions and future perspectives” providing further comments in this section: “In recent years, OCTA has become indispensable in clinical practice for both the diagnosis, follow-up of AMD and to evaluate the response to treatment with anti-VEGF agents. OCTA is a quick and non-invasive diagnostic tool that does not require the use of a contrast agent. It helps to identify early signs of AMD; it shows a good level of sensitivity and accuracy in the diagnosis of late stage of AMD; it also provides the correct characterization of subtypes of MNV improving the detection rate and offering new insights into pathophysiology and progression of these lesions.

OCTA provides some biomarkers as predictors of functional and anatomical outcomes improving patient management and therapy. Moreover, OCT-A has several advantages, including noninvasive and rapid acquisition of angiograms, depth resolution, and perfect microvascular resolution. On the contrary, the main limitation of current OCTA devices are image artifacts; however, OCTA technology is evolving in order to improve image quality, acquisition time and automatic interpretation of scans, allowing for improved in diagnosis and understanding of AMD [125].

Visualization of the choroid and choriocapillaris may be affected by the loss of flow signal with depth in short-wavelength OCT-A devices, but this limitation may be addressed in the next future by new-generation of OCT-A devices. In particular, artifacts of masking and unmasking are very common in patients with AMD. These regions are lined with areas of PEDs or large retinal vessels anterior to the choroid on B scans. The creation of artifacts due to PEDs are accompanied by focal unmasking artifacts due to RPE atrophy. In the region immediately below RPE atrophy, an increase in OCT reflectivity throughout the choroid. This it is also accompanied by augmented artifacts decorrelation signal in the underlying sub-levels.

   Despite these disadvantages, OCT-A is a promising imaging modality that may provide various precious information in patients with AMD, and may help in correlating functional parameters. Moreover OCT-A might provide deeper knowledge in the pathophysiology of human choroid AMD and possibly allows predicting the natural history of the disease and choosing the best therapeutic approach. In particular, noninvasive in vivo identification of the histopathologic substrate and understanding of biogenesis of first signs of the disease will open the door to further therapeutic targets.”

(page 13-14, line 811-873)

Table 1. How were the devices selected to be included in this table? Please mention how was your choice made.

Response: We agree and thank the Reviewer for the comment. The OCTA devices discussed in the paragraph 4 are those for which there are more information in the literature, which have been the subject of previous comparisons and which are most often used for clinical studies. It is difficult to justify this choice on the basis of an objective issue. We modified the text as follows: “Table 1 shows the comparison of different commercially available OCTA devices:” (page 4, line 234)

In material and methods section authors provided search criteria. What are the results of the search, in detail? Please provide this information.

Response: We agree with you and added the following details in the text: “A preliminary Pubmed search strategy was built up by an experienced information specialist, aided by all authors, combining free-text terms and synonyms of: “Optical coherence tomography angiography” and “age-related macular degeneration”. Only scientific articles wrote in English language in the last 5 years were included for internal review and were further evaluated. Of overall 2246 scientific articles, after analyzing manuscript′s titles and abstracts, 89 articles were selected. Moreover, few select articles published before 2015 have been cited for historical purposes.” (page 2-3, line 93-160).

Paragraph 5 and the following paragraphs: I would start the paragraph from listing the most important aspects/problems in OCTA diagnosis of iAMD and than follow with review of literature referring to that. This kind of “pattern” I would suggest to use in all paragraphs: this way you will have a form of a review, not an essay.

Response: We agree with the Reviewer and therefore we modified all text as suggested.

Moderate English editing and spell-check required.

Response: We are sorry for typos and language’s errors. A professional native English scientific writer reviewed the text.

Reviewer 2 Report

The authors summarised a big part of the current state and application of OCTA in intermediate and late AMD, giving a nice and comprehensive overview about opportunities and limitations.

There are some remarks:

The authors give examples of possible limitations and artefacts in OCTA imaging. What about wrong segmentation that occurs when retinal structure is distorted seriously, e.g. due to oedema, pigment epithelial detachment? Please comment.

The wavelength used in SS-OCTA of 1050 nm has the advantage of a deeper penetration into the tissue. The question arises what happens to the possible resolution of imaging compared to SD-OCTA, as it depends on the used wavelength. Please comment.

In Table 1, imaging size is not given in mm x mm, and NOT pixels.

In line 159, the authors write about “3 x 3 or 6 x 6 scans”. Please change to “3 x 3 or 6 x 6 mm² ”.

 Line 264: Do the authors mean OCT or OCTA?

The reviewer was not able to see the “kissing sign” in Figure 3B.

In general, a number of slight grammatical errors and typos should be corrected.

Author Response

Corresponding Author:

Dr Adriano Carnevali, MD

Telephone: +39 09613647041

Email : adrianocarnevali@live.it

Dear Editor,

Thank you for considering our revised manuscript entitled Optical Coherence Tomography Angiography in Intermediate and Late Age-Related Macular Degeneration: Review of Current Technical Aspects and Applicationsfor publication in Applied Sciences.

The points raised by the Reviewers have been favorably taken into account and incorporated into the revised manuscript. We provide a point-by-point response to each of their comments below.

Best regards,

Adriano Carnevali

Comments from the editors and reviewers:

Reviewer #2

The authors summarized a big part of the current state and application of OCTA in intermediate and late AMD, giving a nice and comprehensive overview about opportunities and limitations.

There are some remarks:

The authors give examples of possible limitations and artefacts in OCTA imaging. What about wrong segmentation that occurs when retinal structure is distorted seriously, e.g. due to oedema, pigment epithelial detachment? Please comment.

Response: We agree with you and we modified the section “Conclusions and future perspectives” adding the limitations about the artefacts in OCTA imaging as follows: “Visualization of the choroid and choriocapillaris may be affected by the loss of flow signal with depth in short-wavelength OCT-A devices, but this limitation may be addressed in the next future by new-generation of OCT-A devices. In particular, artifacts of masking and unmasking are very common in patients with AMD. These regions are lined with areas of PEDs or large retinal vessels anterior to the choroid on B scans. The creation of artifacts due to PEDs are accompanied by focal unmasking artifacts due to RPE atrophy. In the region immediately below RPE atrophy, an increase in OCT reflectivity throughout the choroid. This it is also accompanied by augmented artifacts decorrelation signal in the underlying sub-levels.” (page 13, line 824-831)

The wavelength used in SS-OCTA of 1050 nm has the advantage of a deeper penetration into the tissue. The question arises what happens to the possible resolution of imaging compared to SD-OCTA, as it depends on the used wavelength. Please comment.

Response: We agree and thanks the Reviewer for the comment. The increased wavelength allows for better tissue penetration and therefore a better wavelength. We have added modified text as follows: “Since that resolution depends mainly on the wavelength, increasing this parameter allowed a better axial resolution, resulting in a better characterization of the different layers even in the presence of obstacles such as the presence of cataracts or vitreous opacities.” (page 5, line 269-271)

In Table 1, imaging size is not given in mm x mm, and NOT pixels.

Response: Thanks for the comment. We corrected the text accordingly.

In line 159, the authors write about “3 x 3 or 6 x 6 scans”. Please change to “3 x 3 or 6 x 6 mm²”.

Response: Corrected.

Line 264: Do the authors mean OCT or OCTA?

Response: In this paragraph we mean OCT scan.

The reviewer was not able to see the “kissing sign” in Figure 3B.

Response: We agree with you and deleted the sign in question from the legend. The new version of the Figure legend is: ”Optical coherence tomography angiography (OCTA) and structural optical coherence tomography (OCT) of a Type III macular MNV. (A) 3x3 OCTA of outer retinal segment showing presence of a white “point like” that corresponds to a detectable flow. (B) Co-registered structural OCT scan with hyporeflective cyst and hyperreflective intraretinal lesion”. (page 10, line 549-542)

In general, a number of slight grammatical errors and typos should be corrected.

Response: We are sorry for typos and language’s errors. A professional native English scientific writer reviewed the text.

Round 2

Reviewer 1 Report

Good job ! I am looking forward to see this piece in print.

Congrats